# Long-term *in vitro* monitoring of AAV-transduction efficiencies in real-time with Hoechst 33342

**Xiaonan Hu, Roland Meister, Jan Tode ⓘ, Carsten Framme, Heiko Fuchs ⓘ***

Institute of Ophthalmology, University Eye Hospital, Hannover Medical School, Hannover, Germany

* fuchs.heiko@mh-hannover.de

**Data Availability Statement:** All relevant data are within the manuscript and its Supporting Information files.

**Funding:** The author(s) received no specific funding for this work.

## Abstract

Adeno-associated viral transduction allows the introduction of nucleic fragments into cells and is widely used to modulate gene expressions *in vitro* and *in vivo*. It enables the study of genetic functions and disease mechanisms and, more recently, serves as a tool for gene repair. To achieve optimal transduction performance for a given cell type, selecting an appropriate serotype and the number of virus particles per cell, also known as the multiplicity of infection, is critical. Fluorescent proteins are one of the common reporter genes to visualize successfully transduced cells and assess transduction efficiencies. Traditional methods of measuring fluorescence-positive cells are endpoint analysis by flow cytometry or manual counting with a fluorescence microscope. However, the flow cytometry analysis does not allow further measurement in a test run, and manual counting by microscopy is time-consuming. Here, we present a method that repeatedly evaluates transduction efficiencies by adding the DNA-stain Hoechst 33342 during the transduction process combined with a microscope or live-cell imager and microplate image analysis software. The method achieves fast, high-throughput, reproducible, and real-time post-transduction analysis and allows for optimizing transduction parameters and screening for a proper approach.

## Introduction

Transferring nucleotide sequences into cells is a powerful approach to studying gene functions, pathological mechanisms, and therapeutic effects. There are various transfection methods, including physical, chemical, and biological techniques, of which the efficiencies vary greatly depending on plenty of factors [1, 2]. Selecting an appropriate gene delivery system is essential to achieve optimal transfection efficiency and low toxicity. The success and degree of transfection could be directly confirmed by the protein or mRNA expression level of the gene of interest by western blot or quantitative PCR (qPCR) [3–6]. However, multiple experiment factors affect the result, making it less reliable and standardized. Besides, it requires internal control and cannot determine the percentage of successfully transfected cells due to the development being on an overall level of a population of cells [7]. Using reporter genes to analyze transfection efficiency is more straightforward to monitor and can compare different methods and

**Competing interests:** The authors have declared that no competing interests exist.

reagents before introducing the target gene. Among all kinds of reporter genes, fluorescent reporters such as a labeled fluorescent molecule and green fluorescent protein (GFP) enable the visualization of positively transfected cells and the localization of the protein of interest. Flow cytometry is one of the most frequently used methods of determining the transfection efficiency of fluorescent-positive cells [8–15]. It is performed as an endpoint analysis since cells need to be harvested. Therefore, the method is time-consuming, and the transfection efficiency cannot be evaluated before or after this point. Another technique uses fluorescence microscopy and image software like ImageJ to count fluorescent positive cells [16, 17]. Although the analysis can be done repeatedly, evaluating many variants takes time and effort.

In this study, we report a method of determining the transduction efficiency in real-time with Adeno-associated virus (AAV) coding for eGFP using an automated fluorescent microscope and analysis software. For accurate cell counting, Hoechst 33342 (Hoechst) was added to the cells before the transduction. Hoechst is a DNA-binding fluorochrome that can stain the fluorescent blue nuclei in both live and dead cells. It is widely used in endpoint analysis at a high concentration due to its binding with the DNA groove [18]. However, according to our previous work, a concentration between 7 to 28 nM is neither cytotoxic nor inhibits cellular proliferation and provides a sufficient staining of cell nuclei. In addition, the acquisition parameters were optimized to exclude possible phototoxic effects due to the excitation light [19]. This method is efficient as it provides real-time transduction rates due to its arbitrary repeatability and is, therefore, suitable for high-throughput analyses.

## Materials and methods

### Cell isolation and culture

Human retinal pigment epithelium (hRPE) cells and human scleral fibroblasts (hSF) were isolated from patients undergoing enucleation. The procedures were carried out following the tenets of the Declaration of Helsinki, with the consent of the ethics committee of Hannover Medical School. An informed written agreement was obtained from all subjects. Patient samples were collected during the period August 2021 and May 2023. The enucleated eyeball was disinfected in 70% ethanol and cut into halves. Scleral tissues were excised, cut into small pieces, and cultured in a 6-well plate. Cell outgrowth was observed after approximately one week. At about 90% confluence, cell passages were performed. The isolation procedure of hRPE cells was based on our previous study [20]. The anterior segment, lens, and vitreous body were removed. The RPE sheet was peeled off and digested with TrypLE™ Express Enzyme (Gibco #12604021, Waltham, MA, U.S.) on a thermo-shaker. The detached RPE cells were collected and resuspended in a 6-well plate with a complete medium. When hRPE cells reached around 70–90% confluency, they were passaged. Both cells of passages 3–6 were used for experiments.

A complete medium was composed of DMEM/F-12 medium (Gibco #21331–020), 10% fetal bovine serum (Pan-Biotech #P40-39500, Aidenbach, Germany), 1% GlutaMAX™ (Gibco #3505–061), and 1% Penicillin-Streptomycin (Gibco #15140–122).

### Hoechst preparation

Hoechst (Sigma-Aldrich B2261-25MG) was dissolved in sterile double distilled water to a concentration of 1 mg/ml and sterile-filtered using a 33 mm diameter PES syringe filter with a pore size of 0.22 μm. The solution was aliquoted and stored at -20˚C. In experiments, the solution was diluted in sterile double distilled water or medium and added directly into the cell medium.

## Adeno-associated virus transduction

$1 \times 10^4$ hRPE cells and the spontaneously arising RPE cell line, ARPE-19 cells, were seeded in each well of 48-well plates with 250 μl complete medium, respectively. $2.5 \times 10^3$ hSFs were seeded in each well of 96-well plates with 100 μl complete medium. Four ng/$10^4$ cells of Hoechst, which corresponds to a concentration of 28 nM that can be used in live cells for long-term staining, according to our previous study [19], were added to stain the nuclei. For hRPE cells, self-complementary AAV-2-CMV-GFP, AAV-6-CMV-GFP, and AAV-DJ-CMV-GFP particles purchased from Charles River (Wilmington, MA, U.S.) were suspended in the same medium and added into cells at a multiplicity of infection (MOI) of $5 \times 10^4$ GC/cell (the following units are identical). The plates were then incubated at 37˚C. The medium was completely changed every 3 days. 28 nM Hoechst was added every five days. For ARPE-19 cells, AAV-DJ-CMV-GFP particles were added into cells at different MOIs of $5 \times 10^4$, $10 \times 10^4$, and $20 \times 10^4$. The plate was monitored continuously in the live-cell imager (BioTek® Lionheart™ FX automated microscope) for 4 days. For hSFs, AAV-2-CMV-GFP particles were added at different MOIs of $2.5 \times 10^4$, $5 \times 10^4$, and $10 \times 10^4$. The plate was incubated in the live-cell imager for 5 days.

## Imaging

Two hours after transduction, when Hoechst was uptake by the cells, the first fluorescent images were taken in the 48-well culture plate. A single image was taken from each well using a 4x objective or multiple images using a 10x objective with an inverted microscope. Extending the range to include the maximum number of cells is recommended. For this purpose, montage images per well were taken using a live-cell imager with an automated XYZ stage function and later stitched together. Aside from the phase contrast (PC) channel, a blue fluorescence filter of an excitation/emission wavelength of 377/447 nm was used to detect Hoechst-stained nuclei and a green filter of a 469/525 nm wavelength was used to detect the GFP signal.

## Image analysis

**Image stitching.**   If several pictures were taken in each well, "Image Stitching" was performed to fuse separate montage images into one. The images were processed and analyzed with microplate analysis software. Gen5 Image Prime 3.05 (Santa Clara, CA, U.S.) was used in our case.

**Background processing.**   Background processing was applied in the Hoechst and GFP channels in the stitched images. The parameters were adjusted to reduce background noise without obviously affecting fluorescent signals. Here, we set "Background" to "Dark" and auto "Background flattening" with a "Rolling Ball diameter" of 650 μm and "Image smoothing strength" of "0".

**Cellular analysis.**   After background subtraction, all Hoechst-stained nuclei in the image were gated to count total cell numbers. For the Hoechst channel, "Background" was set to "Dark" and "Threshold value" of primary masks' intensity to 5000. "Object size" was set between 10 and 50 μm with the options of "Split touching objects", "Fill holes in masks", and "Analyze entire image" selected. Under "Advanced Detection Options", "Rolling Ball diameter" was adjusted to 10 μm, "Image smoothing strength" to "0", and the background was evaluated on 5% of the lowest pixels for further background flattening. The above parameters should be tuned according to respective experiments, cell types, and analysis software to gate the nuclei as much as possible. Metrics of "Cell Count", "Object Peak GFP" (the peak GFP intensity in primary masks), and "Object Mean GFP" (the mean GFP intensity in primary masks) were calculated.

**Determination of the threshold of GFP-positive fluorescence.** The GFP intensity in the first images (taken 2 hours after transduction) was considered the baseline or the background noise. The highest GFP intensity of all the Hoechst-stained cells in the whole plate was designated the GFP-positive threshold. We used both "Object Peak GFP" and "Object Mean GFP" metrics to make the standard more stringent. For instance, 2 hours after transduction, the highest "Object Peak GFP" intensity was 1900, and the highest "Object Mean GFP" intensity was 750 on our plate. Therefore, the condition of "Object Peak GFP" over 1900 and "Object Mean GFP" over 750 was set as the threshold to identify GFP-positive cells.

**Subpopulation analysis and transduction efficiency calculation.** Within the primary masks gated at the previous step, objects with a GFP intensity higher than the set condition were labeled as GFP-positive cells, successfully transduced cells. The cell numbers of both primary gated objects and sub-populated objects were calculated. The transduction efficiency was defined as the number of GFP-positive cells (subpopulation objects) divided by the number of Hoechst-positive nuclei (primary masks).

The image acquisition and analysis parameters can be saved as a protocol, allowing the transduction efficiency to be automatically analyzed repeatedly.

## Automated count

The automated analysis was performed with Gen5 software. For background processing, the background was set to "Light", "Rolling Ball diameter" to 96 μm, and "Image smoothing strength" to 8. In cellular analysis, the "Threshold" was configured to 2000 with the object size between 25–140 μm. "Analyze entire image", "Fill holes in masks" and "Split touching objects" were checked. In advanced detection options, "Rolling Ball diameter" was set to 50 μm, "Image smoothing strength" to 0, and the background was evaluated on 5% of the lowest pixels.

## Comparative analysis with ImageJ

The Hoechst and GFP fluorescent images of hRPE cells transduced with AAV-2-CMV-GFP at 24 hours from technical triplicates were analyzed with ImageJ separately. The images were first converted to grayscale (8-bit) under the toolbar "Image > Type". Under "Adjust > Threshold", the particles with a brightness threshold between 10 and 200 were separated from the background. "Fill Holes" and "Watershed" under "Process > Binary" are applied when necessary. The selected particles were counted with "Analyze > Analyze Particles".

## Statistical analysis

All AAV serotypes were tested in at least three biological replicates, and each biological replicate was tested with at least three technical replicates at each run. The data were analyzed with GraphPad Prism 9 (Boston, MA, U.S.). An unpaired two-tailed t-test was performed to detect significant differences between the data from ImageJ and our proposed method. A *p*-value $< 0.05$ was statistically significant.

## Results

First, we would like to briefly present our *in vitro* method for determining transduction efficiency in real-time by adding Hoechst. The summarized process is shown in Fig 1. Nuclei were stained with Hoechst before or simultaneously with transduction to allow for more accurate cell counting. The first image was taken approximately 2 hours later as Hoechst was efficiently uptaken to establish the protocol. While a single image per well was sufficient with a low-magnification objective, it was recommended that multiple images be taken with a high-

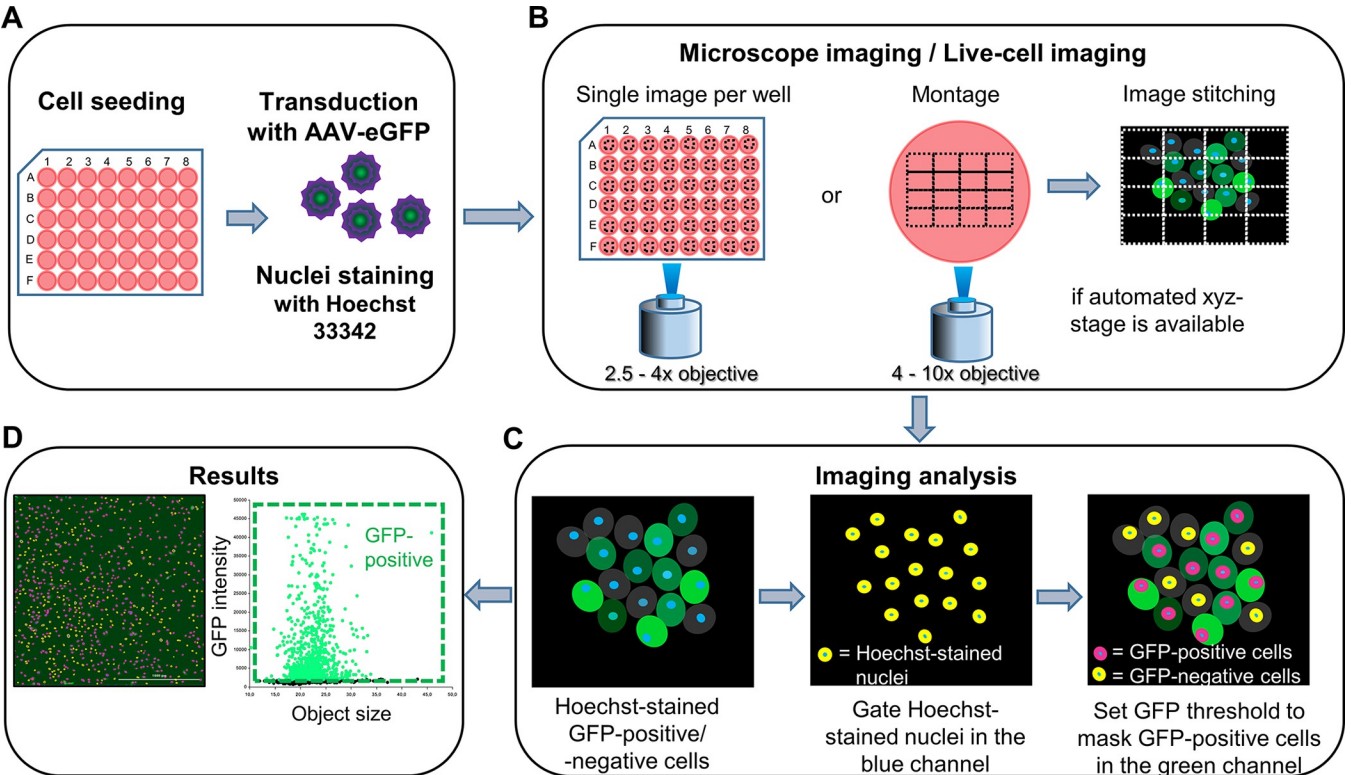

**Fig 1. The walkthrough.** (A) Transduction was performed, and the live DNA dye, Hoechst, was added after seeding cells in a 48-well plate. (B) After 2 hours, when all nuclei were efficiently stained with Hoechst, fluorescent images were taken with an inverted microscope or a live-cell imager if possible. For each well, one image with a lower objective or a stitched image composed of several images with a higher objective was taken. (C) Images were processed and analyzed with microplate analysis software. Hoechst-stained nuclei were gated as primary objects for total cell number counting in the blue channel. A GFP threshold was determined, and the primary mask was used to sub-gate GFP-positive cells in the green channel. The transduction efficiency was defined as the percentage of GFP-positive cells. (D) Representative results of analyzed Hoechst and GFP images 24 hours after AAV-2-CMV-GFP transduction in hRPE cells. Yellow-labeled objects were GFP-negative cells, and red-labeled objects were GFP-positive cells. The scale bar represented 1000 μm.

magnification objective and stitched together to include as many cells as possible. Background processing of the acquired image to reduce signal noise was optional and should be adjusted based on individual experiments. Hoechst-stained nuclei were primarily gated for cellular analysis using the Hoechst signal and object size. The threshold for GFP-positive cells was determined by the highest and mean background GFP signal intensity and utilized for the sub-population of GFP-positive cells. This analysis could be repeated as GFP-fluorescent cells appeared. The numbers of Hoechst-stained nuclei and GFP-positive cells were automatically calculated, and transduction efficiency was defined as the percentage of GFP-positive cells.

Secondly, we validated the feasibility of our method by transducing hRPE cells with AAV-2-CMV-GFP. After 24 hours of transduction, we observed successful and extensive transduction in the captured multi-channel fluorescence image (Fig 2A). The image was then processed using Gen5 software with the following steps: reduction of background signal noise and adjustment of gating parameters for primary masks. As a result, 1281 Hoechst-stained nuclei were automatically counted as the total number of cells in the blue channel. In addition, by setting the GFP-positive threshold, which was calculated before cells exhibited GFP fluorescence, we identified 813 GFP-positive cells with intensities above the threshold. To provide a more intuitive visualization of the extent of transduction, we generated a scatter plot of GFP intensity versus object size from three technical replicates, which indicated a successful transduction rate of 63.5% (Fig 2B and 2C). In addition, the images were analyzed with ImageJ software (National

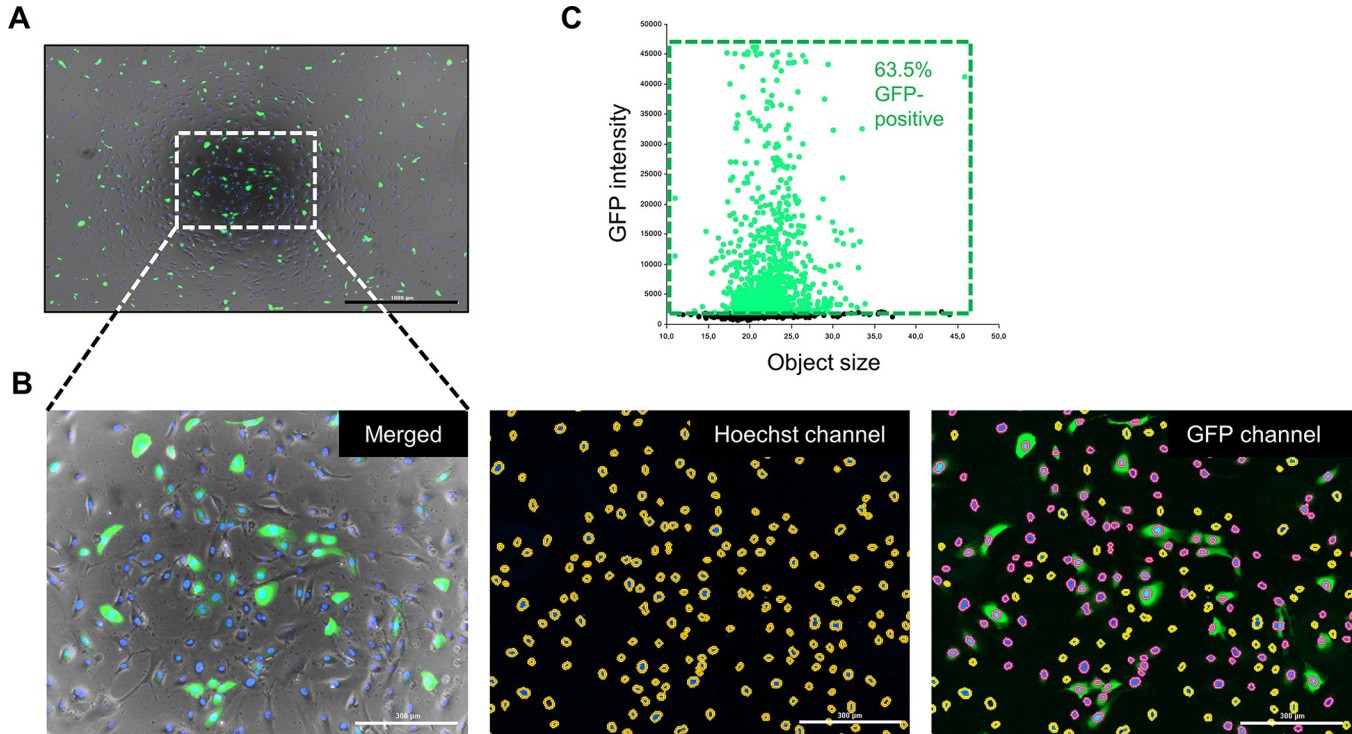

**Fig 2. Example results.** (A) A representative stitched and merged PC, Hoechst, and GFP fluorescence image 24 hours after hRPE cells were transduced with AAV-2-CMV-GFP. The scale bar of 1000 μm was represented. (B) The magnified insert from the box outlined in (A). Hoechst-stained nuclei in the blue channel were masked in yellow. Among them, GFP-positive cells were labeled in red in the green channel. The scale bars represented 300 μm. (C) The scatter plot of GFP intensity versus the object size of three technical replicates.

Institutes of Health, Bethesda, MD, https://ij.imjoy.io/) to validate the accuracy of our method. The results showed no significant difference in transduction efficiency with our proposed method (S1 Fig).

Next, we explored the various features and advantages of our method. Using a 48-well plate and the automated calculation process allowed us to analyze and compare several transduction factors simultaneously. For example, we can evaluate different transduction serotypes, MOIs, cell types, and cell numbers in a single run. First, we compared different serotypes in the same cell type by transducing AAV-2-CMV-GFP, AAV-6-CMV-GFP, and AAV-DJ-CMV-GFP at the same MOI in hRPE cells and monitored them for 14 days. At 24 hours, the respective cell counts in one replicate of each group were 1281, 1169, and 1384, of which 813 (63.5%), 363 (31.1%), and 803 (58.0%) were GFP-positive. After 72 hours, the number of GFP-positive cells increased to 1543 (99.4%), 1450 (56.4%), and 2479 (97.4%) in a total of 1553, 2570, and 2546 cells from one replicate in each group (Fig 3A). The scatter plot of three technical replicates showed transduction rates of 99.4%, 59.7%, and 97.6% for AAV-2, AAV-6, and AAV-DJ at 72 hours (Fig 3B). Furthermore, the transduction efficiency of AAV-2 and AAV-DJ remained consistently higher than that of AAV-6, peaking at 48 hours and lasting for 7 days, while AAV-6 showed lower efficiency and shorter transduction duration (Fig 3C).

Second, we compared different MOIs of the same serotype by transducing the ARPE-19 cells with AAV-DJ-CMV-GFP at MOIs of zero, 5 x$10^4$, 10 x$10^4$, and 20 x$10^4$. In order to show a coherent process demonstrating the benefits of a live-cell imager, the plate was incubated in the imager and monitored continuously every 30 minutes for 96 hours. At 48 hours, the cell numbers in each group were 1404, 944, 671, and 366, with GFP-positive cells being 0 (0%), 725

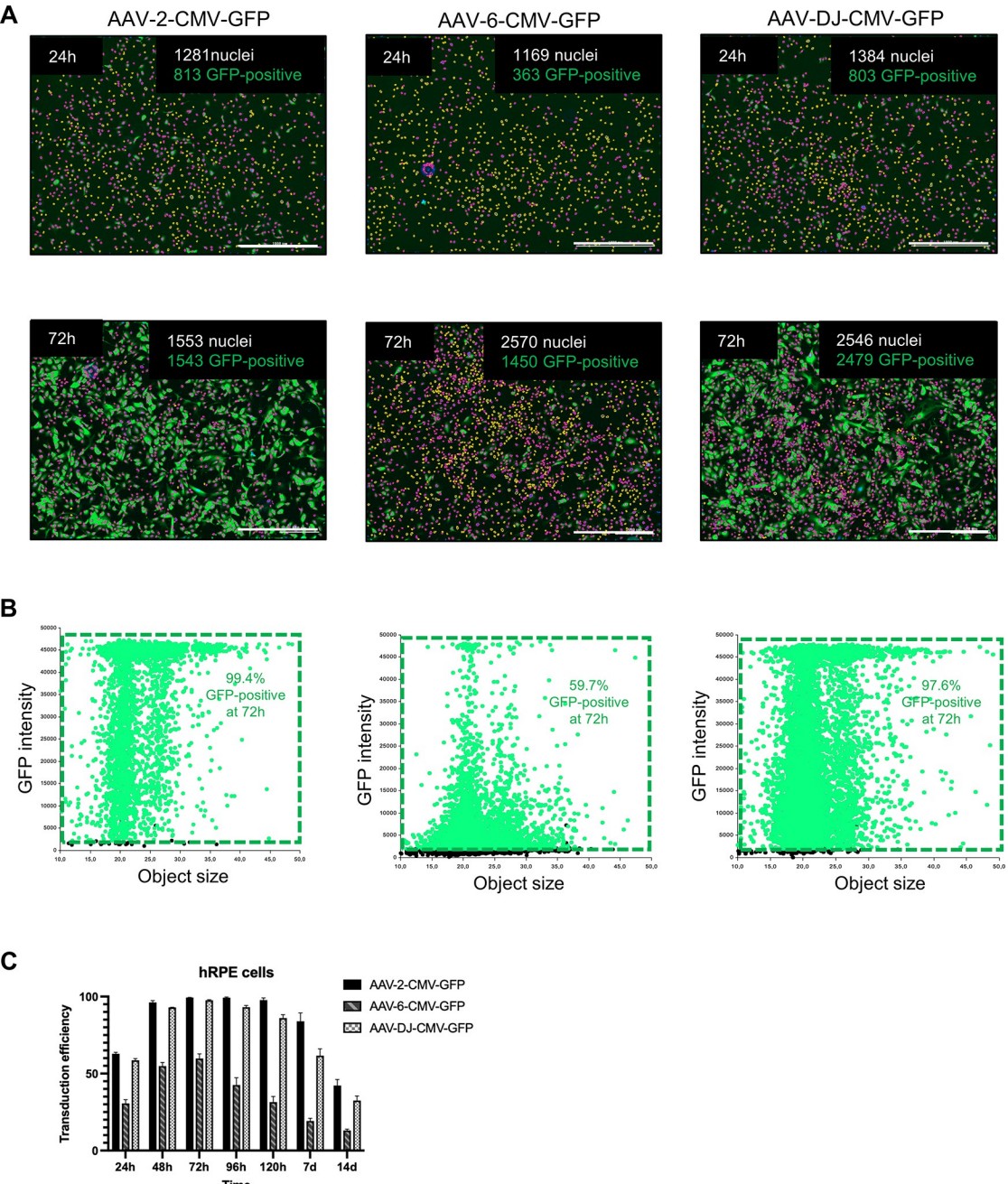

**Fig 3. 14-day long-term determination of GFP-positive hRPE cells transduced with AAV-2-CMV-GFP, AAV-6-CMV-GFP, or AAV-DJ-CMV-GFP.** The transduction was performed using 1 x $10^4$ hRPE cells with AAV-2, AAV-6, or AAV-DJ at an MOI of 5 x $10^4$, and the cells were stained with 4 ng/$10^4$ cells of Hoechst. (A) Representative analyzed Hoechst and GFP fluorescence images 24 hours and 72 hours after transduction. Hoechst-stained nuclei, labeled in yellow, indicated GFP-negative cells, while those marked in red indicated GFP-positive cells. The scale bars of 1000 μm were used. (B) Scatter plots of GFP intensity versus object size from three technical replicates. (C) The transduction efficiencies of AAV-2, AAV-6, and AAV-DJ at different time points within 14 days from three biological replicates with four technical replicates each.

(76.8%), 661 (98.5%), and 358 (97.8%) respectively (Fig 4A). The corresponding scatter plot of six technical replicates showed transduction rates of 0%, 78.5%, 98.5%, and 98.3% (Fig 4B). Although higher efficiencies were achieved at higher MOIs, cell proliferation and cell viability

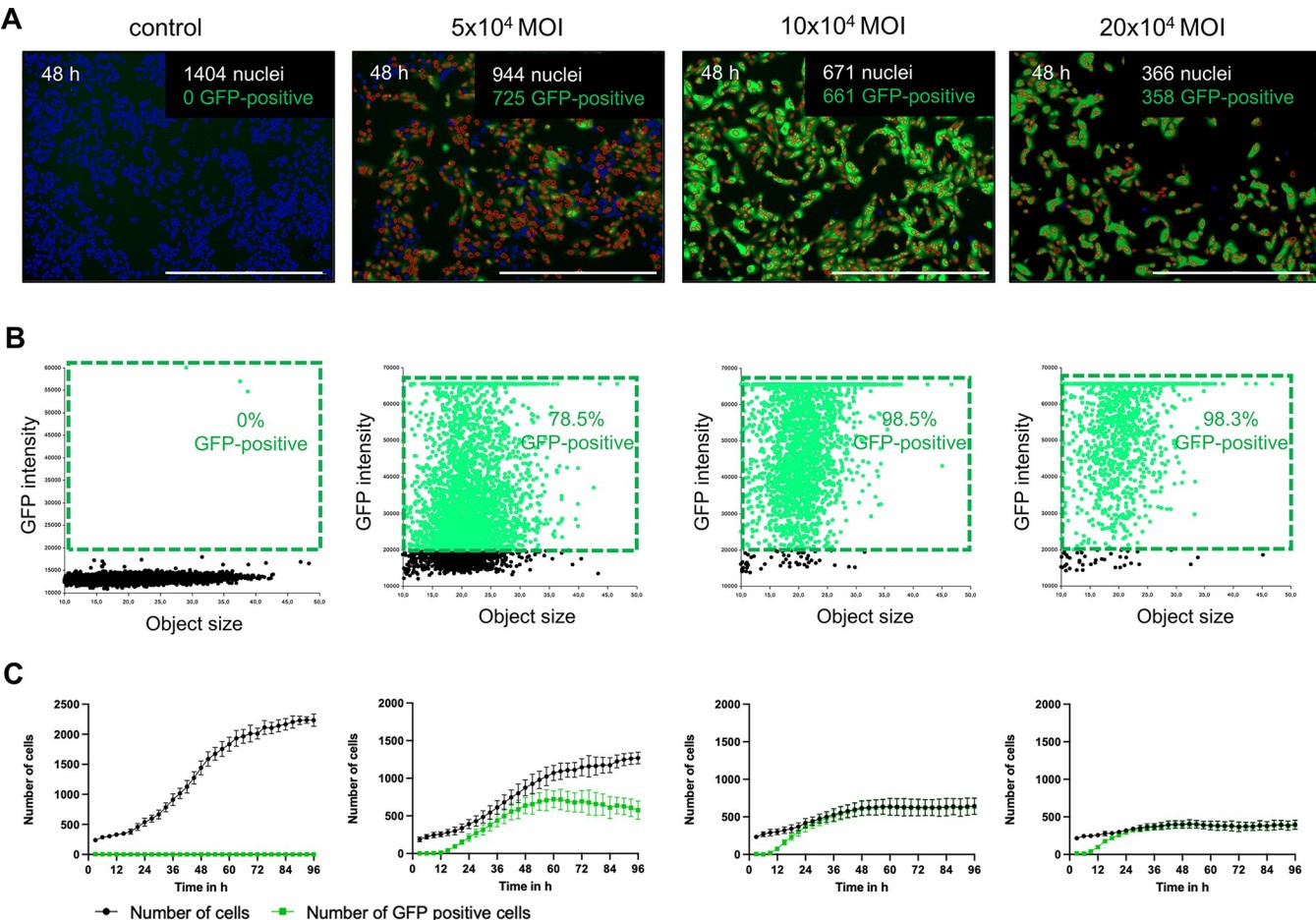

**Fig 4. Transduction of ARPE-19 cells with AAV-DJ-CMV-GFP at different MOIs.** 1 x10⁴ ARPE-19 cells were transduced with AAV-DJ using different MOIs of 0, 5 x10⁴, 10 x10⁴, and 20 x10⁴ and stained with 4 ng/10⁴ cells Hoechst. Live-cell imaging was performed for 96 h consecutively. (A) Representative analyzed Hoechst and GFP fluorescence images 48 hours after transduction. Hoechst-stained nuclei labeled in blue indicated GFP-negative, and the ones marked in red indicated GFP-positive cells. The scale bars represented 1000 μm. (B) Scatter plots of GFP intensity versus object size of six technical replicates. (C) The numbers of Hoechst-stained cells (black) and GFP-positive cells (green) at different MOIs over 96 hours.

were significantly inhibited (Fig 4C). A supplemental time-lapse video of GFP fluorescence in ARPE-19 cells after AAV-DJ transduction over 96 hours is provided (S1 Video).

Since the hRPE and ARPE-19 cells used in this study are of epithelial origin and are characterized by their round morphology and almost uniform size, automatic gating can provide results nearly identical to Hoechst gating (S2 Fig). Finally, to emphasize the advantages of our method, we chose human sclera fibroblasts (hSF), which vary considerably in size and shape. We performed the transduction with AAV-2-CMV-GFP at MOIs of zero, 2.5 x10⁴, 5 x10⁴, and 10 x10⁴ in 96-well format, often used for high-throughput analyses and monitored continuously for 120 hours (Fig 5).

## Discussion

Selecting an appropriate method and finding optimal transduction conditions is critical for subsequent formal experiments. The most straightforward way of determining the transduction efficiency is testing the expression level, either protein or mRNA level, of the gene of interest by western blot, ELISA, or qPCR. However, such methods require specific antibodies or

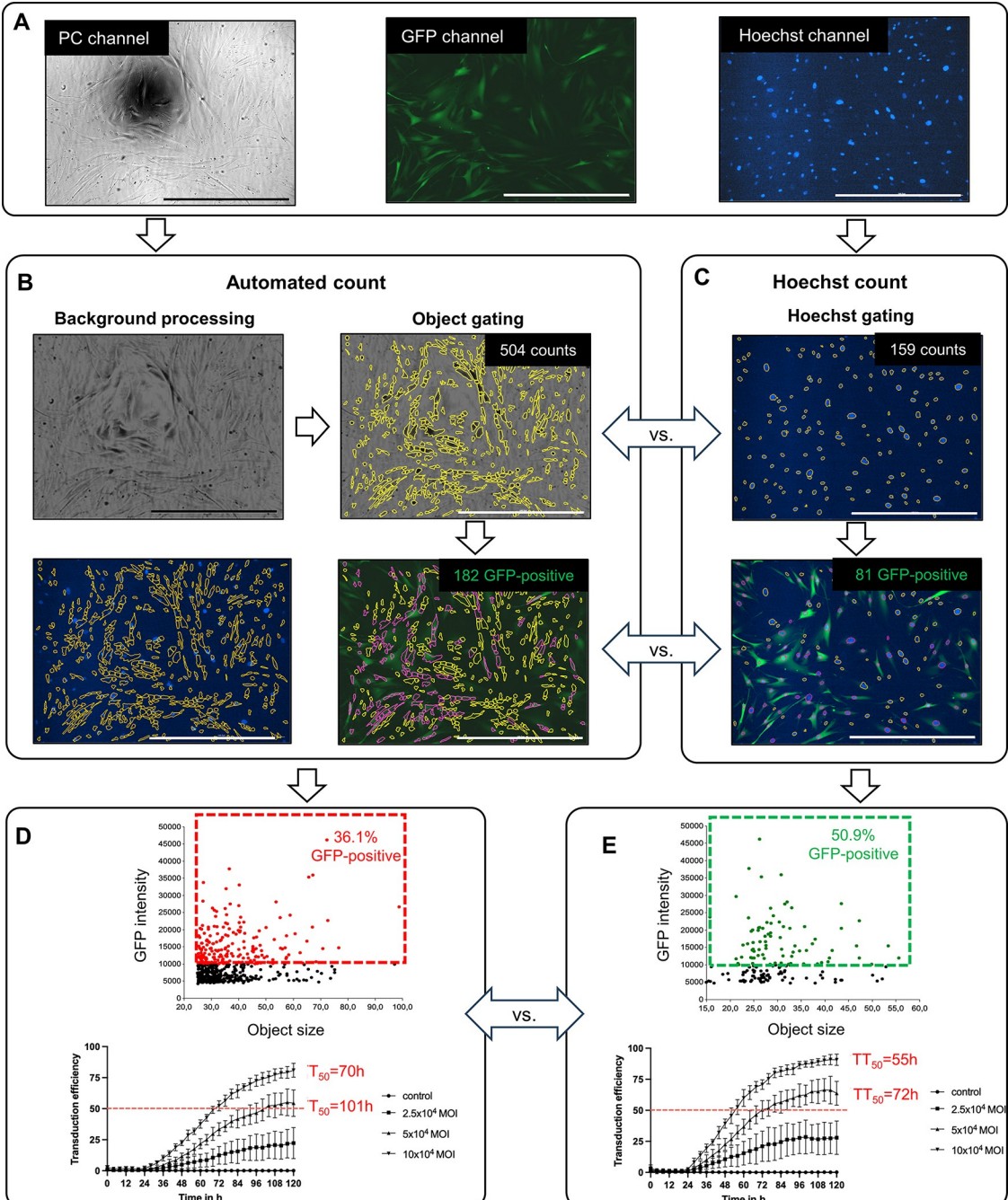

**Fig 5. High-throughput analysis of hSFs cultivated in a 96-well format transduced with AAV-2-CMV-GFP at different MOIs.**
$2.5 \times 10^3$ hSFs were transduced with AAV-2 using different MOIs of 0, $2.5 \times 10^4$, $5 \times 10^4$, and $10 \times 10^4$ in a 96-well plate, and the nuclei were stained with 4 ng/$10^4$ cells Hoechst. Live-cell imaging was performed for 120 h consecutively. (A) Representative raw images of PC, GFP, and Hoechst channels 73 hours after 5 x104 MOI AAV-2 transduction. (B) Automated count analysis. Upper left: PC image after background processing. Upper right: primary masks are gating dark objects (yellow) on a light background in the background-processed image. Lower right: submasked GFP-positive objects (red) among the primary masks in the GFP channel picture. Lower left: primary masks in the Hoechst channel picture. (C) Cell nuclei stained with Hoechst. Upper: primary masks of cell nuclei (yellow) in the Hoechst channel. Lower: Submasked GFP-positive objects (red) are among the primary masks in the merged Hoechst and GFP channels picture. (D) Upper: the scatter plot of GFP intensity versus the object size. Lower: the transduction efficiencies of AAV-2 at different MOIs over 5 days. The data were collected from eight technical replicates and analyzed with the automated count. (E) Upper: the scatter plot of GFP intensity versus the object size. Lower: the transduction efficiencies of AAV-2 at different MOIs over 5 days. The data were collected from eight technical replicates and analyzed with Hoechst count. The scale bars represent 1000 μm.

primers and cost more time [7]. Additionally, each working step might lead to different results, making them unsuitable for preliminary screening of a proper transduction reagent. To better search for an optimized transduction method, reporter genes are applied to standardize transduction efficiencies among various forms and conditions. The most commonly used ones are fluorescent molecules, luciferase, or β-galactosidase [17, 21, 22]. Among these, fluorescent molecules are easy to monitor and visualize and can be used to localize targets. Flow cytometry or microscopy is commonly used to measure the percentage of fluorescent-positive cells. The primary issue with flow cytometry is that it only allows endpoint analysis and is time-consuming. Many materials are needed to acquire information on transduction efficiencies at multiple time points and conditions. Manual cell counting with a microscope is labor-intensive and impracticable when testing many variants. In this study, we try to address these shortcomings by combining a microscope and microplate analysis software to present a new protocol for measuring fluorescent transduction efficiencies over time. Since AAV vectors are one of the established transduction approaches applied in diseases related to the eye, liver, muscle, and nervous system and have numerous serotypes with different affinities to cells, we chose AAV vectors to transduce cells to exhibit the advantages of our method [1, 2, 23–25]. We recommend using the DNA live dye Hoechst 33342 to stain the nuclei for accurate cell counting. The Hoechst mask can subsequently be used to identify GFP-positive cells in the GFP channel.

In previous work, we have shown that by optimizing the acquisition parameters, a low Hoechst concentration in the medium is sufficient to visualize the nuclei over five days without significantly affecting cell activities such as cell proliferation or signaling cascades. Although the appropriate Hoechst concentration varies from cell type to cell type, a concentration of 7 to 28 nM is recommended based on our previous study [19]. Usually, cell bodies are gated for automated counting when nuclei are not stained. However, the meniscus effect reduces the contrast between cells and the background at the periphery in PC images and is more pronounced in smaller diameter wells such as 48 and 96-well plates (Fig 5A). Due to the overexposure at the outer edges of the image, cells are hardly recognizable. They cannot be optically separated from the background, making an additional background processing step necessary, which requires additional computing time and storage capacity. Even after background processing, it is difficult to detect the cell bodies depending on the cell type (Fig 5B). For example, cells with an irregular body shape, such as fibroblast-like cells, are much more challenging to be correctly gated. Parameters used for image preprocessing and gating also vary between users, leading to user-dependent results. In Fig 5, we compared automated and Hoechst counts on hSF cultivated in a 96-well format. With automated count, cells are challenging to recognize separately, and the object gating of cell bodies is unreliable, showing a significant discrepancy with the Hoechst count of the nuclei. 504 events were identified with automated cell counting, whereas only 159 nuclei were counted in the Hoechst channel. The sub-mask of the automated cell count revealed 182 GFP-positive hSFs in the GFP channel, while the Hoechst count identified only 81 GFP-positive hSFs (Fig 5B and 5C).

In the lower left picture of panel B, many primary masks generated from automated cell counting included no nuclei. Since the automated cell count for hSF is imprecise, the percentage of GFP-positive cells and the calculated transduction time 50 ($TT_{50}$) are incorrect compared to the Hoechst-gated nuclei (Fig 5D and 5E). With the assistance of Hoechst, fluorescent nuclei, which are usually more regular in size and spatially separated from each other, appear more prominent in the background and can be gated more efficiently. Considering that Hoechst also stains dead cells, other dyes that specifically stain dead cells or debris, such as propidium iodide (PI), can be applied simultaneously, as demonstrated in our prior publication [19]. PI-positive cells are detected with a red fluorescence filter and sub-masked within the primary Hoechst-stained objects. The number of viable cells is calculated by subtracting the

number of PI-positive nuclei from Hoechst-positive nuclei. The transduction efficiency can be measured using the number of viable cells instead of total Hoechst-positive nuclei for a more accurate evaluation.

Nevertheless, cytotoxicity does not necessarily lead to the death of cells but can also manifest itself through reduced viability, like a reduced proliferation rate. One advantage of our method is that not only the transduction rates but also the proliferation rates can be determined simultaneously. For example, ARPE-19 transduced with AAV-DJ-CMV-GFP at an MOI of only $5x10^4$ vg/cell show a significant inhibition of proliferation and, thus, impairment of their cell viability (Fig 4C).

In addition to the advantage that our method allows to monitor cell proliferation and transduction rates in one run, our method has the following advantages.

First, it is feasible and fast. Our approach can be used either with an inverted fluorescence microscope or a live-cell imager combined with microplate analysis software. There is no need to use flow cytometry or more sophisticated instruments. It is cost-effective and practicable in all laboratories equipped with an inverted fluorescence microscope. Once the primary parameters have been set and saved, the entire analysis process can be automated to save time.

Second, it is efficient and allows real-time, high-throughput transduction analysis. It can assess multiple variables at the same time, such as transduction methods, reagent concentrations, cell types, and cell densities. Therefore, the efficiencies of various viral serotypes with different affinities to other cell types can be evaluated much faster.

Third, it is repeatable and delivers real-time results. It can be performed anytime without fixing or harvesting the cells. Alternatively, if a live-cell imager is available, continuous monitoring for several days can be performed, and the data can be analyzed at more intense intervals in this case. With a real-time measurement of transduction efficiencies, our method can provide researchers with more information, such as the time needed to reach the peak efficiency and the most extended duration of the transduction.

Last but not least, the error of our method is negligible. Single or multiple images covering as many cells as possible are suggested for each well. Also, manual counting is avoided to reduce random errors. Besides, using Hoechst-stained nuclei for cell counting is more reliable than the counting method using PC pictures.

However, this method is not suitable for cells that are not at the same focal level, for example, suspension cells and organoids. It is trivial, but it should be mentioned that cell counting with Hoechst only works with mononuclear cells. Cells with a high nucleus-to-cytoplasm ratio, such as embryonic stem cells, in which the nuclei may be too close together to be gated separately, also reach the limits of this method. In addition, the Hoechst concentration for specific cell types should be tested beforehand to ensure that the cell nuclei visually stand out from the background.

In general, our method can facilitate future studies in quickly screening out suitable transduction methods and conditions, offering a clue for optimizing experiments and speeding up experimental progress.

## Supporting information

**S1 Video. A time-lapse video of GFP fluorescence in ARPE-19 cells after the transduction of AAV-DJ-CMV-GFP at different MOIs over 96 hours.**
(MP4)

**S1 Fig. Transduction of hRPE cells with AAV-2-CMV-GFP after 24 hours.** The fluorescent images were analyzed with ImageJ. (A) Representative raw and analyzed images of Hoechst and GFP channels. The scale bars of 1000 μm were represented. (B) Transduction efficiencies

of three technical replicates were calculated with ImageJ and our proposed method. Ns, non-significant.
(TIF)

**S2 Fig. Comparison between Hoechst and automatic cell counting in ARPE-19 cells.** The top picture shows PC image of the Hoechst-gated nuclei (marked in yellow). The bottom picture shows the PC image after background subtraction with automated gated objects (highlighted in yellow). The scale bar represents 1000 μm.
(TIF)

**S1 Table. Raw data set for Figs 3–5, SF 1+2.**
(XLSX)

## Author Contributions

**Conceptualization:** Xiaonan Hu.

**Data curation:** Xiaonan Hu, Roland Meister.

**Formal analysis:** Xiaonan Hu, Roland Meister, Heiko Fuchs.

**Methodology:** Xiaonan Hu.

**Project administration:** Heiko Fuchs.

**Supervision:** Jan Tode, Carsten Framme, Heiko Fuchs.

**Writing – original draft:** Xiaonan Hu.

**Writing – review & editing:** Xiaonan Hu, Roland Meister, Jan Tode, Carsten Framme, Heiko Fuchs.

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
