## [Decision Letter · Decision Letter 0]

16 Oct 2023

PONE-D-23-28137Long-term in vitro monitoring of AAV-transduction efficiencies in real-time with Hoechst 33342PLOS ONE

Dear Dr. Fuchs,

Thank you for submitting your manuscript to PLOS ONE. After careful consideration, we feel that it has merit but does not fully meet PLOS ONE’s publication criteria as it currently stands. Therefore, we invite you to submit a revised version of the manuscript that addresses the points raised during the review process.

We look forward to receiving your revised manuscript.

Kind regards,

Chen Ling, Ph.D.

Academic Editor

PLOS ONE

Reviewers' comments:

Reviewer's Responses to Questions

**Comments to the Author**

1. Is the manuscript technically sound, and do the data support the conclusions?

Reviewer #1: Yes

Reviewer #2: Yes

Reviewer #3: Yes

2. Has the statistical analysis been performed appropriately and rigorously? 

Reviewer #1: Yes

Reviewer #2: Yes

Reviewer #3: Yes

3. Have the authors made all data underlying the findings in their manuscript fully available?

Reviewer #1: Yes

Reviewer #2: Yes

Reviewer #3: Yes

4. Is the manuscript presented in an intelligible fashion and written in standard English?

Reviewer #1: Yes

Reviewer #2: Yes

Reviewer #3: Yes

5. Review Comments to the Author

Reviewer #1: In this highly interesting manuscript, the authors describe a novel method for in vitro monitoring of AAV transduction, by using GFP transduction and Hoechst 33342 staining. The authors present a very useful method with the possibility of serial measurements. All results are sound and the conclusions drawn are convincing. The linguistic style is good, which makes the manuscript easy to read and understand. The work presented will be of high interest to your readers. Therefore I strongly recommend publication, whithout further changes to the manuscript necessary.

Reviewer #2: This study employed a simple protocol combining Hst33342 and a transgene reporter (GFP) to track AAV transduction efficiency during 2D cell culture. This involved quantifying imaged cells in both primary human RPE and RPE cell lines. This procedure equipped by microscope and image analysis with live cell and GFP positive cell counting would be straightforward.

While the gathered data is intriguing, there is room for improvement in the study's design to better elucidate its novelty and feasibility.

Reviewer suggested a major revision with following comments and questions.

Major Revisions:

1. Clarify whether hst33342 is added one-time at seeding or recharged during long-term culture with medium replacement.

2. line 308: In addition to utilize hst33342 as a live cell marker, with transgene GFP, which serves as a visible marker for monitoring AAV transduction, author need to collect more data to show the benefit or superiority of Hst33342 application. Additionally, using Gen5 Image Prime 3.05 (Santa Clara, CA, U.S.) software for image analysis may be less valuable if it still in market, compared to other free and open-source software like Image J. Author need to state the equipment and software can be adopted to universal microscope imager.

3. While the procedure with hst33342 to monitor AAV transduction has been developed using human primary cell RPEs, it seems suitable only for 2D adhered cell culture. Please discuss the applicability of this method to other primary cells in suspension or 3D conditions.

4. line 312: For high throughput, consider using 96 or 384 well plates instead of the 48 well plate format. Additionally, explore the possibility of automated or programmable analysis for real-time monitoring of AAV transduction. Reducing the number of cells in 96/384 plate could also streamline data analysis.

5. To achieve long-term monitoring of AAV transduction, consider extending the duration beyond 4 days post-AAV infection. The authors might also contemplate transitioning this method from cell culture to ex vivo or in vivo models for more meaningful insights.

6. For cell types with multiple nuclei, such as primary human hepatocytes, integrating intelligent software programs to outline cell borders could enhance the significance of this work.

Minor Revisions:

1. Fig1A: please use another icon to represent AAV virions.

2. For each figure, please indicate each color.

3. Line 71: Could you please provide the full name of "hRPE" at its first mention in the manuscript? The same applies to "ARPE" later in the text.

4. It is recommended to enhance the quality of almost all figures to ensure they are easily recognizable and comprehensible.

5. line 149: considering the GFP intensity not saturated, the exposure time also need to be optimized for each batch of AAV transduction.

6. The reviewer is interested in whether the signal from hst33342 or the GFP signal in the image-based analysis can be correlated with the readings from the plate reader, which is less time-consuming.

Reviewer #3: In this manuscript, the authors propose a method for real-time detection of AAV transfection efficiency by pre-treating cells with Hoechst 33342, a DNA fluorescent dye, before transfection. The treated cells were then analyzed using specialized instruments for real-time monitoring. The authors experimentally validated the feasibility of this method and compared it with traditional methods.

Main Suggestions:

The introduction section lacks sufficient information about Hoechst 33342. Please provide a detailed explanation of Hoechst 33342 in the Introduction to offer readers a comprehensive understanding of its role in the experiment.

Clarify how the minimal dosage of Hoechst 33342, used as a nuclear stain, was determined in this experiment. Provide the rationale or methodology behind selecting this specific dosage.

The manuscript only mentions the use of RPE cells. Explain the reason for choosing this specific cell line. However, the authors did not investigate the effectiveness of this method on other cell types, making the conclusion overly optimistic. It is recommended to justify the choice of ocular cells and provide a background on current AAV research and treatments related to ocular diseases. Relevant studies such as PMID: 37273779 and PMID: 35408825 can be cited to support the rationale.

The manuscript lacks a seamless integration between the methods and images. Many instances merely state that the method can be applied without immediate validation from experimental data. To enhance the manuscript’s credibility, include experimental data comparisons between this method and traditional approaches. This will allow readers to visualize the advantages of this method.

Provide a detailed discussion on how the accuracy of this method was validated. Readers require comprehensive insights into the validation process, including methodologies and results, to fully comprehend the reliability of the proposed technique.

6. PLOS authors have the option to publish the peer review history of their article (what does this mean?). If published, this will include your full peer review and any attached files.

Reviewer #1: **Yes: **Michael C Burger

Reviewer #2: **Yes: **Chengwen Li

Reviewer #3: **Yes: **Chen Ling

---

## [Author Response · Author response to Decision Letter 0]

27 Nov 2023

Dear reviewers,

First, we thank the reviewers for taking the time to read and review our manuscript. We have improved the manuscript based on their comments and suggestions and would like to address their points in detail below.

Responses to the Reviewer #1 comments.

In this highly interesting manuscript, the authors describe a novel method for in vitro monitoring of AAV transduction, by using GFP transduction and Hoechst 33342 staining. The authors present a very useful method with the possibility of serial measurements. All results are sound and the conclusions drawn are convincing. The linguistic style is good, which makes the manuscript easy to read and understand. The work presented will be of high interest to your readers. Therefore I strongly recommend publication, without further changes to the manuscript necessary.

Answer: 

We thank the reviewer for the positive comment.

Responses to the Reviewer #2 comments.

This study employed a simple protocol combining Hst33342 and a transgene reporter (GFP) to track AAV transduction efficiency during 2D cell culture. This involved quantifying imaged cells in both primary human RPE and RPE cell lines. This procedure equipped by microscope and image analysis with live cell and GFP positive cell counting would be straightforward. While the gathered data is intriguing, there is room for improvement in the study's design to better elucidate its novelty and feasibility.

Reviewer suggested a major revision with following comments and questions.

Major Revisions:

1. Clarify whether hst33342 is added one-time at seeding or recharged during long-term culture with medium replacement.

Answer: 

1. 28 nM Hoechst 33342 was added every five days. We clarified this in the method (Line 107): " The medium was changed every 3 days. 28 nM Hoechst was added every five days."

2. The dosage of Hoechst was revised due to a calculation mistake. 4 ng/104 cells, instead of 6 ng/104 cells, Hoechst corresponds to 28 nM (Line 101): "Four ng/104 cells of Hoechst, which corresponds to a concentration of 28 nM that can be used in live cells for long-term staining, according to our previous study [19], were added to stain the nuclei."

2. line 308: In addition to utilize hst33342 as a live cell marker, with transgene GFP, which serves as a visible marker for monitoring AAV transduction, author need to collect more data to show the benefit or superiority of Hst33342 application. Additionally, using Gen5 Image Prime 3.05 (Santa Clara, CA, U.S.) software for image analysis may be less valuable if it still in market, compared to other free and open-source software like Image J. Author need to state the equipment and software can be adopted to universal microscope imager.

Answer: 

1. We added two figures to compare Hoechst cell counting and normal phase contrast counting. We elaborated on the superiority of Hoechst cell counting in the discussion (Line 371): "Usually, cell bodies are gated for automated counting when nuclei are not stained. However, the meniscus effect reduces the contrast between cells and the background at the periphery in PC images and is more pronounced in smaller diameter wells such as 48 and 96-well plates (Fig 5A). Due to the overexposure at the outer edges of the image, cells are hardly recognizable. They cannot be optically separated from the background, making an additional background processing step necessary, which requires additional computing time and storage capacity. Even after background processing, it is difficult to detect the cell bodies depending on the cell type (Fig 5B). For example, cells with an irregular body shape, such as fibroblast-like cells, are much more challenging to be correctly gated. Parameters used for image preprocessing and gating also vary between users, leading to user-dependent results. In Fig 5, we compared automated and Hoechst counts on hSF cultivated in a 96-well format. With automated count, cells are challenging to recognize separately, and the object gating of cell bodies is unreliable, showing a significant discrepancy with the Hoechst count of the nuclei. 504 events were identified with automated cell counting, whereas only 159 nuclei were counted in the Hoechst channel. The sub-mask of the automated cell count revealed 182 GFP-positive hSFs in the GFP channel, while the Hoechst count identified only 81 GFP-positive hSFs (Fig 5B+C). In the lower left picture of panel B, many primary masks generated from automated cell counting included no nuclei. Since the automated cell count for hSF is imprecise, the percentage of GFP-positive cells and the calculated transduction time 50 (TT50) are incorrect compared to the Hoechst-gated nuclei. (Fig 5D+E). With the assistance of Hoechst, fluorescent nuclei, which are usually more regular in size and spatially separated from each other, appear more prominent in the background and can be gated more efficiently."

2. This method can be adopted with different microscopes and software apart from Gen5, which is only an example in this study (Line 398): "Our approach can be used either with an inverted fluorescence microscope or a live-cell imager combined with microplate analysis software." Additionally, we tested some of the images using ImageJ and compared the results with our method. Line 249: "In addition, the images were analyzed with ImageJ software (National Institutes of Health, Bethesda, MD, https://ij.imjoy.io/) to validate the accuracy of our method. The results showed no significant difference in transduction efficiency with our proposed method (S1 Fig)." Unfortunately, it is unrealistic for us to test it with all software in market. However, the principle and workflow of gating nuclei, determining GFP signal intensity, and calculating GFP-positive cells should work with most analysis software. This study aims to offer a new idea for monitoring transduction efficiency in real time in live cells using Hoechst.

3. While the procedure with hst33342 to monitor AAV transduction has been developed using human primary cell RPEs, it seems suitable only for 2D adhered cell culture. Please discuss the applicability of this method to other primary cells in suspension or 3D conditions.

Answer:

Unfortunately, this method is unsuitable for cells in suspension or 3D conditions. The limitation was clarified in the discussion already (Line 427): "However, this method is not suitable for cells that are not at the same focal level, for example, suspension cells and organoids."

4. line 312: For high throughput, consider using 96 or 384 well plates instead of the 48 well plate format. Additionally, explore the possibility of automated or programmable analysis for real-time monitoring of AAV transduction. Reducing the number of cells in 96/384 plate could also streamline data analysis.

Answer:

1. We added an experiment of transducing human scleral fibroblasts with AAV2 in a 96-well plate for 5 days to verify that this method can be applied in other plate formats (Line 315): "Finally, to emphasize the advantages of our method, we chose hSFs, which vary considerably in size and shape. We performed the transduction with AAV2-CMV-GFP at MOIs of zero, 2.5 x104, 5 x104, and 10 x104 in 96-well format, often used for high-throughput analyses and monitored continuously for 120 hours (Fig 5)."

2. Multi-well plates such as 48 and 96-well for high-throughput analysis are used. However, due to the meniscus effect, cell bodies become less prominent and overexposed at the periphery in smaller diameter wells (Fig 5A). This effect makes it difficult to use automated phase contrast to count cell numbers even after a background subtraction step. Also, automated cell gating and counting are unreliable for irregular cell shapes. In the lower-left picture in Fig 5B, many primary masks do not include nuclei inside, proving to be unreliable. The discussion (Line 371) and the previous comment explained the inferiority of automated count.

5. To achieve long-term monitoring of AAV transduction, consider extending the duration beyond 4 days post-AAV infection. The authors might also contemplate transitioning this method from cell culture to ex vivo or in vivo models for more meaningful insights.

Answer:

1. Although we only presented images at certain time points, the transduction experiment in hRPE lasted for 14 days (Line 266): "First, we compared different serotypes in the same cell type by transducing AAV2-CMV-GFP, AAV6-CMV-GFP, and AAVDJ-CMV-GFP at the same MOI in hRPE cells and monitored them for 14 days." In Fig 3, the tendency of transduction efficiencies over two weeks was shown in a bar chart.

2. One of the limitations is that this method cannot be adopted in organoids, either ex vivo or in vivo models, which was clarified in the discussion (Line 427): "However, this method is not suitable for cells that are not at the same focal level, for example, suspension cells and organoids."

6. For cell types with multiple nuclei, such as primary human hepatocytes, integrating intelligent software programs to outline cell borders could enhance the significance of this work.

Answer:

1. Due to the meniscus effect and depending on the cell type, it is challenging to outline the cell border in multi-well plates even after background processing. There is a great deal of subjectivity and error in this process. However, Hoechst can stain the nuclei and make cell counting more manageable and objective because fluorescent nuclei are prominent in the dark background and separate from each other. The disadvantages of automated count and the superiority of Hoechst count were elaborated in the discussion (Line 371) and the previous comment.

2. This is a limitation of our method that it cannot be adopted in multi-nuclei cells and we added this in the discussion (Line 428): "It is trivial, but it should be mentioned that cell counting with Hoechst only works with mononuclear cells." 

Minor Revisions:

1. Fig1A: please use another icon to represent AAV virions.

Answer:

The icon of AAV was changed in Fig 1.

2. For each figure, please indicate each color.

Answer:

In each figure or figure legend, the meaningful color was already clearly indicated, e.g.,"Hoechst-stained nuclei labeled in blue indicated GFP-negative, and the ones marked in red indicated GFP-positive cells." Other colors are used for aesthetic purposes only, which in the opinion of the authors, are not necessary.

3. Line 71: Could you please provide the full name of "hRPE" at its first mention in the manuscript? The same applies to "ARPE" later in the text.

Answer:

The definitions of the terms were added at the first mention (Line 76 and 98): "Human retinal pigment epithelium (hRPE) cells and human scleral fibroblasts (hSF) were isolated from patients undergoing enucleation." and "1 x104 hRPE cells and the spontaneously arising RPE cell line, ARPE-19 cells, were seeded ".

4. It is recommended to enhance the quality of almost all figures to ensure they are easily recognizable and comprehensible.

Answer:

Due to technical reasons, the sharpness of all figures declined after submitting them to the online system. Nevertheless, a high resolution picture can be downloaded by clicking the link above the Figures in the PDF document.

5. line 149: considering the GFP intensity not saturated, the exposure time also need to be optimized for each batch of AAV transduction.

Answer:

Although the GFP signal is not saturated at the beginning, the exposure time can be set relatively longer to ensure a sensitive detection of GFP signal. The images' backgrounds can be subtracted afterward in analysis. Also, the histogram of each channel can be adjusted as long as the same capture parameters are applied. Nevertheless, a pretest can be considered to figure out an optimized acquisition time.

6. The reviewer is interested in whether the signal from hst33342 or the GFP signal in the image-based analysis can be correlated with the readings from the plate reader, which is less time-consuming.

Answer:

Plate readers equipped with an integrated inverted microscope, which can record individual cells and nuclei, are suitable for our method. Plate readers that do not have this feature can only determine the average intensity for Hoechst and GFP but cannot provide any information about the transduction efficiency.

Responses to the Reviewer #3 comments

In this manuscript, the authors propose a method for real-time detection of AAV transfection efficiency by pre-treating cells with Hoechst 33342, a DNA fluorescent dye, before transfection. The treated cells were then analyzed using specialized instruments for real-time monitoring. The authors experimentally validated the feasibility of this method and compared it with traditional methods.

Main Suggestions:

The introduction section lacks sufficient information about Hoechst 33342. Please provide a detailed explanation of Hoechst 33342 in the Introduction to offer readers a comprehensive understanding of its role in the experiment.

Answer: 

A more detailed introduction of Hoechst 33342 was added (Line 62): "For accurate cell counting, Hoechst 33342 (Hoechst) was added to the cells before the transduction. Hoechst is a DNA-binding fluorochrome that can stain the fluorescent blue nuclei in both live and dead cells. It is widely used in endpoint analysis at a high concentration due to its binding with the DNA groove [18]. However, according to our previous work, a concentration between 7 to 28 nM is neither cytotoxic nor inhibits cellular proliferation and provides a sufficient staining of cell nuclei. In addition, the acquisition parameters were optimized to exclude possible phototoxic effects due to the excitation light [19]."

Clarify how the minimal dosage of Hoechst 33342, used as a nuclear stain, was determined in this experiment. Provide the rationale or methodology behind selecting this specific dosage.

Answer: 

1. The reason of the dosage of Hoechst was added in the discussion (Line 366): "In previous work, we have shown that by optimizing the acquisition parameters, a low Hoechst concentration in the medium is sufficient to visualize the nuclei over five days without significantly affecting cell activities such as cell proliferation or signaling cascades. Although the appropriate Hoechst concentration varies from cell type to cell type, a concentration of 7 to 28 nM is recommended based on our previous study [19]." To show a stronger contrast of the nuclei, 28 nM Hoechst was applied.

2. The dosage of Hoechst was revised due to a calculation mistake. 4 ng/104 cells, instead of 6 ng/104 cells, Hoechst corresponds to 28 nM (Line 101): "Four ng/104 cells of Hoechst, which corresponds to a concentration of 28 nM that can be used in live cells for long-term staining, according to our previous study [19], were added to stain the nuclei."

The manuscript only mentions the use of RPE cells. Explain the reason for choosing this specific cell line. However, the authors did not investigate the effectiveness of this method on other cell types, making the conclusion overly optimistic. It is recommended to justify the choice of ocular cells and provide a background on current AAV research and treatments related to ocular diseases. Relevant studies such as PMID: 37273779 and PMID: 35408825 can be cited to support the rationale.

Answer: 

1. AAV transduction is a widely used approach in various disciplines focusing on all kinds of cells, including ocular cells. The choice of ocular cells is based on the nature of our laboratory. We used both human RPE cells and the cell line ARPE-19 to indicate that this method can be applied in both primary cells and cell lines.

2. We added an experiment of transducing AAV2 in human scleral fibroblasts to show that this method can be used in other cell type (Line 315): "Finally, to emphasize the advantages of our method, we chose hSFs, which vary considerably in size and shape. We performed the transduction with AAV2-CMV-GFP at MOIs of zero, 2.5 x104, 5 x104, and 10 x104 in 96-well format, often used for high-throughput analyses and monitored continuously for 120 hours (Fig 5)." However, it is unrealistic to validate this method in all cell types. The purpose of our study is to offer a new idea to monitoring transduction efficiency in real-time in live cells with the assistance of Hoechst.

The manuscript lacks a seamless integration between the methods and images. Many instances merely state that the method can be applied without immediate validation from experimental data. To enhance the manuscript's credibility, include experimental data comparisons between this method and traditional approaches. This will allow readers to visualize the advantages of this method.

Answer: 

1. To validate the accuracy, we used ImageJ to analyze the images of hRPE cells transduced with AAV2 after 24 h. The representative pictures were shown in S1 Fig. A t-test indicated that the transduction efficiencies calculated with two methods had no significant difference. Line 249: "In addition, the images were analyzed with ImageJ software (National Institutes of Health, Bethesda, MD, https://ij.imjoy.io/) to validate the accuracy of our method. The results showed no significant difference in transduction efficiency with our proposed method (S1 Fig)." However, there is no gold standard so far. ImageJ is only a commonly used method.

2. In addition, nuclei counting after fluorescent staining is relatively reliable today. In Fig 5 and S2 Fig, we compared traditional automated count and Hoechst count. In phase contrast images, meniscus effect causes it difficult to recognize cell bodies and requires additional background subtraction. However, the processing is user-dependent and cell type-dependent. The outline of cell borders is not trustworthy even after background processing. The drawback of traditional automated counting and the reliability of Hoechst counting was elaborated in the discussion (Line 371): "Usually, cell bodies are gated for automated counting when nuclei are not stained. However, the meniscus effect reduces the contrast between cells and the background at the periphery in PC images and is more pronounced in smaller diameter wells such as 48 and 96-well plates (Fig 5A). Due to the overexposure at the outer edges of the image, cells are hardly recognizable. They cannot be optically separated from the background, making an additional background processing step necessary, which requires additional computing time and storage capacity. Even after background processing, it is difficult to detect the cell bodies depending on the cell type (Fig 5B). For example, cells with an irregular body shape, such as fibroblast-like cells, are much more challenging to be correctly gated. Parameters used for image preprocessing and gating also vary between users, leading to user-dependent results. In Fig 5, we compared automated and Hoechst counts on hSF cultivated in a 96-well format. With automated count, cells are challenging to recognize separately, and the object gating of cell bodies is unreliable, showing a significant discrepancy with the Hoechst count of the nuclei. 504 events were identified with automated cell counting, whereas only 159 nuclei were counted in the Hoechst channel. The sub-mask of the automated cell count revealed 182 GFP-positive hSFs in the GFP channel, while the Hoechst count identified only 81 GFP-positive hSFs (Fig 5B+C). In the lower left picture of panel B, many primary masks generated from automated cell counting included no nuclei. Since the automated cell count for hSF is imprecise, the percentage of GFP-positive cells and the calculated transduction time 50 (TT50) are incorrect compared to the Hoechst-gated nuclei. (Fig 5D+E). With the assistance of Hoechst, fluorescent nuclei, which are usually more regular in size and spatially separated from each other, appear more prominent in the background and can be gated more efficiently."

Provide a detailed discussion on how the accuracy of this method was validated. Readers require comprehensive insights into the validation process, including methodologies and results, to fully comprehend the reliability of the proposed technique.

Answer:

We added an analysis of the images of hRPE cells transduced with AAV2 after 24 h using ImageJ (Line 249): "In addition, the images were analyzed with ImageJ software (National Institutes of Health, Bethesda, MD, https://ij.imjoy.io/) to validate the accuracy of our method. The results showed no significant difference in transduction efficiency with our proposed method (S1 Fig)." Nevertheless, there is no gold standard to validate the accuracy of our method today. The superiority of nuclei counting with Hoechst was elaborated in the discussion (Line 371) and in the previous comment.

---

## [Decision Letter · Decision Letter 1]

18 Dec 2023

PONE-D-23-28137R1Long-term in vitro monitoring of AAV-transduction efficiencies in real-time with Hoechst 33342PLOS ONE

Dear Dr. Fuchs,

Thank you for submitting your manuscript to PLOS ONE. After careful consideration, we feel that it has merit but does not fully meet PLOS ONE’s publication criteria as it currently stands. Therefore, we invite you to submit a revised version of the manuscript that addresses the points raised during the review process.

We look forward to receiving your revised manuscript.

Kind regards,

Chen Ling, Ph.D.

Academic Editor

PLOS ONE

Journal Requirements:

Reviewers' comments:

Reviewer's Responses to Questions

**Comments to the Author**

1. If the authors have adequately addressed your comments raised in a previous round of review and you feel that this manuscript is now acceptable for publication, you may indicate that here to bypass the “Comments to the Author” section, enter your conflict of interest statement in the “Confidential to Editor” section, and submit your "Accept" recommendation.

Reviewer #2: All comments have been addressed

Reviewer #3: (No Response)

2. Is the manuscript technically sound, and do the data support the conclusions?

Reviewer #2: Yes

Reviewer #3: Yes

3. Has the statistical analysis been performed appropriately and rigorously? 

Reviewer #2: Yes

Reviewer #3: Yes

4. Have the authors made all data underlying the findings in their manuscript fully available?

Reviewer #2: Yes

Reviewer #3: Yes

5. Is the manuscript presented in an intelligible fashion and written in standard English?

Reviewer #2: Yes

Reviewer #3: Yes

6. Review Comments to the Author

Reviewer #2: There are notable concerns regarding the accurate differentiation between dying or dead cells and debris from live cells in images without manual evaluation. However, the addition of a single drop of propidium iodide (PI) enables the exclusion of debris or dead cells in flow cytometry. At present, there seems to be no compelling advantage over flow cytometry. Even with the capacity of a 96T plate reader, detecting fewer cells from various time points under identical settings is feasible. Image capture and analysis might require a longer operation and analysis compared to flow cytometry.

Line 101: It is imperative to provide the product information of Hst 33342, including details about the solvent, storage conditions, and shelf life for the execution of this work.

Line 103: We typically utilize AAV-D/J or AAV-DJ instead of AAVDJ.

Line 104: Please specify the AAV type as ss or sc?

Line 106: Our standard practice involves using the Multiplicity of Infection (MOI) as vg/cell for AAV application in vitro.

Line 108: Was the medium completely replaced at 100%?

Reviewer #3: In general, the author has addressed the feedback effectively by providing additional information, justifications, and experimental data where necessary. The manuscript appears to have been improved based on the feedback received.

Other concerns are as follows:

Adding more literature will make the manuscript more comprehensive. The author did not cite the correct literature. (refer to previous review comments 3)

7. PLOS authors have the option to publish the peer review history of their article (what does this mean?). If published, this will include your full peer review and any attached files.

Reviewer #2: **Yes: **Chengwen Li

Reviewer #3: **Yes: **Chen Ling

---

## [Author Response · Author response to Decision Letter 1]

4 Jan 2024

Dear reviewers,

We appreciate your comments and made necessary adjustments based on the suggestions. We would like to address the points in detail below.

Responses to Reviewer #2 comments.

There are notable concerns regarding the accurate differentiation between dying or dead cells and debris from live cells in images without manual evaluation. However, the addition of a single drop of propidium iodide (PI) enables the exclusion of debris or dead cells in flow cytometry. At present, there seems to be no compelling advantage over flow cytometry. Even with the capacity of a 96T plate reader, detecting fewer cells from various time points under identical settings is feasible. Image capture and analysis might require a longer operation and analysis compared to flow cytometry. 

Answer: 

1. PI is a dead-cell dye that is not only used in flow cytometry. PI can be also dopted in our method to differentiate dead cells or debris, as demonstrated in our prior publication. Hoechst-stained objects are well suited to be used as primary masks in a red channel, allowing the identification of PI-positive cells. The detection and calculation is the same as GFP-positive cells. The detailed method is added in the discussion (Line 409): "Considering that Hoechst also stains dead cells, other dyes that specifically stain dead cells or debris, such as propidium iodide (PI), can be applied simultaneously, as demonstrated in our prior publication [19]. PI-positive cells are detected with a red fluorescence filter and sub-masked within the primary Hoechst-stained objects. The number of viable cells is calculated by subtracting the number of PI-positive nuclei from Hoechst-positive nuclei. The transduction efficiency can be measured using the number of viable cells instead of total Hoechst-positive nuclei for a more accurate evaluation."

2. Manual evaluation is impractical in cases of abundant cell numbers, irregular cell shapes, and colonies. In Fig 5, the phase contrast image and its processed images indicate that manual counting is unsuitable. The disadvantages have already been clarified in the discussion (Line 382).

3. After repeating the method in a 96-well plate as requested in the first revision, we are sorry that you don’t see the advantages over flow cytometry. The disadvantages of flow cytometry were addressed in the manuscript, namely that it is an endpoint analysis that is time-consuming and materials-consuming (Line 363). In contrast, these are precisely the advantages of our method. Although they were clarified in the discussion already, we would like to emphasize again:

1) Our method is not limited to an endpoint analysis. It allows transduction efficiencies to be repeatedly measured over several days at any desired time point. (Line 439)

2) Using a 48- or 96-well plate allows evaluating and monitoring multiple transduction parameters in one run, such as viral serotypes, MOIs, cell types and cell numbers. (Line 434)

3) For example, for a plate tested at 30-minute intervals for 72 hours, 145-time points per well can be analyzed in our method. However, using flow cytometry, 145 experiments are needed to determine these 145 time points. At your request, we have repeated our analysis in 96-well format. How many wells would be needed to monitor 96 conditions in 30-minute intervals for 72 h by flow cytometry? Exactly 96 x 145 wells.

4) Even if labeling dead cells (e.g. with PI) makes sense in some cases, transduction with AAV can impair the viability of cells. Nevertheless, a reduced cell viability does not necessarily lead to cell death but rather to a reduced proliferation rate, as shown in Fig 4, where we tested different MOIs. Can you also determine the proliferation rates in real-time with flow cytometry, as we have shown with our method? Therefore, we included in Line 417:“Nevertheless, cytotoxicity does not necessarily lead to the death of cells but can also manifest itself through reduced viability, like a reduced proliferation rate. One advantage of our method is that not only the transduction rates but also the proliferation rates can be determined simultaneously. For example, ARPE-19 transduced with AAV-DJ-CMV-GFP at an MOI of only 5x104 vg/cell show a significant inhibition of proliferation and, thus, impairment of their cell viability (Fig. 4 C).”

5) As the manuscript outlines, utilizing multiple images per well allows for a more detailed analysis, contributing to a comprehensive understanding of the experimental results. However, it is worth noting that there are no significant differences in transduction efficiencies, whether analyzing 3000 or “only” 1000 cells.

6) Compared to flow cytometry, our method requires less time and less materials, including fewer cells, fewer cell media, less viral particles, and therefore is more resource-efficient and user-friendly.

4. It is fast to perform one run because the analysis is used as an automated program protocol. With a more sophisticated machine, such as a live-cell imager, image capture is also automated, and it takes around 30 minutes to test a 48-well plate. In addition, our method eliminates the need to harvest cells and prepare samples.

1. Line 101: It is imperative to provide the product information of Hst 33342, including details about the solvent, storage conditions, and shelf life for the execution of this work.

Answer: 

The corresponding information of Hoechst 33342 was added in Line 98: “Hoechst (Sigma-Aldrich B2261-25MG) was dissolved in sterile double distilled water to a concentration of 1 mg/ml and sterile-filtered using a 33 mm diameter PES syringe filter with a pore size of 0.22 µm. The solution was aliquoted and stored at -20°C. In experiments, the solution was diluted in sterile double distilled water or medium and added directly into the cell medium.”

2. Line 103: We typically utilize AAV-D/J or AAV-DJ instead of AAVDJ.

Answer: 

The names of AAV in the manuscript and figures were changed. For example, from “AAVDJ” to “AAV-DJ”.

3. Line 104: Please specify the AAV type as ss or sc?

Answer:

In our study, we used scAAV to transduce cells and have included this information in the method section. However, we provide a better approach to measure transduction efficiencies, and the choice of viral vectors depends on the researcher. 

4. Line 106: Our standard practice involves using the Multiplicity of Infection (MOI) as vg/cell for AAV application in vitro.

Answer:

The unit was added in Line 114: “For hRPE cells, AAV-2-CMV-GFP, AAV-6-CMV-GFP, and AAV-DJ-CMV-GFP particles purchased from Charles River (Wilmington, MA, U.S.) were suspended in the same medium and added into cells at a multiplicity of infection (MOI) of 5 x104 GC/cell (the following units are identical).” The following MOI units were omitted since the unit of MOI is normally omitted.

5. Line 108: Was the medium completely replaced at 100%?

Answer:

Yes, the medium was replaced entirely. This statement was added as required. Line 115: “The medium was completely changed every 3 days.”

Responses to Reviewer #3 comments

In general, the author has addressed the feedback effectively by providing additional information, justifications, and experimental data where necessary. The manuscript appears to have been improved based on the feedback received. 

Other concerns are as follows:

Adding more literature will make the manuscript more comprehensive. The author did not cite the correct literature. (refer to previous review comments 3) 

Answer: 

As previously explained, our study aims to offer a new idea for monitoring transduction efficiency in real-time in live cells using Hoechst. Therefore, the choice of cell types and viral vectors depends on the researcher, and the method is not limited to ocular cells and AAV. It is pointless and weird to emphasize current AAV research in ophthalmology because it was not our focus. Nevertheless, as you requested, we have included your previously published article (PMID: 37273779) and the other article you mentioned (PMID: 35408825) in the reference list, stating in Line 370: “Since AAV vectors are one of the established transduction approaches applied in diseases related to the eye, liver, muscle, and nervous system and have numerous serotypes with different affinities to cells, we chose AAV vectors to transduce cells to exhibit the advantages of our method [1,2,23–25].”

We hope that our revised version will be published in PlosOne and remain with kind regards.

Heiko Fuchs

Corresponding Author

---

## [Decision Letter · Decision Letter 2]

22 Jan 2024

Long-term in vitro monitoring of AAV-transduction efficiencies in real-time with Hoechst 33342

PONE-D-23-28137R2

Dear Dr. Fuchs,

We’re pleased to inform you that your manuscript has been judged scientifically suitable for publication and will be formally accepted for publication once it meets all outstanding technical requirements.

Kind regards,

Chen Ling, Ph.D.

Academic Editor

PLOS ONE

Additional Editor Comments (optional):

Reviewers' comments:

Reviewer's Responses to Questions

**Comments to the Author**

1. If the authors have adequately addressed your comments raised in a previous round of review and you feel that this manuscript is now acceptable for publication, you may indicate that here to bypass the “Comments to the Author” section, enter your conflict of interest statement in the “Confidential to Editor” section, and submit your "Accept" recommendation.

Reviewer #2: All comments have been addressed

Reviewer #3: All comments have been addressed

2. Is the manuscript technically sound, and do the data support the conclusions?

Reviewer #2: Yes

Reviewer #3: Yes

3. Has the statistical analysis been performed appropriately and rigorously? 

Reviewer #2: Yes

Reviewer #3: Yes

4. Have the authors made all data underlying the findings in their manuscript fully available?

Reviewer #2: Yes

Reviewer #3: Yes

5. Is the manuscript presented in an intelligible fashion and written in standard English?

Reviewer #2: Yes

Reviewer #3: Yes

6. Review Comments to the Author

Reviewer #2: The author has made additional edits to the content, addressing all previously raised concerns. Regarding the Hst33342 live cell image method with Hst33342 developed by this group for many years, the reviewer strongly recommends that the author could include a comprehensive protocol with step-by-step instructions in the supplementary methods section. This not only enhances the manuscript's quality but also aligns with the standards of PLOS ONE, ensuring that the research is freely accessible to all.

Reviewer #3: (No Response)

7. PLOS authors have the option to publish the peer review history of their article (what does this mean?). If published, this will include your full peer review and any attached files.

Reviewer #2: **Yes: **Chengwen Li

Reviewer #3: No

---

## [Editor Report · Acceptance letter]

21 Feb 2024

PONE-D-23-28137R2 

PLOS ONE

Dear Dr. Fuchs, 

I'm pleased to inform you that your manuscript has been deemed suitable for publication in PLOS ONE. Congratulations! Your manuscript is now being handed over to our production team.

Kind regards, 

on behalf of

Dr. Chen Ling 

Academic Editor

PLOS ONE